# Numerical Optimization of CNT Distribution in Functionally Graded CNT-Reinforced Composite Beams

**DOI:** 10.3390/polym14204418

**Published:** 2022-10-19

**Authors:** J.R. Cho, H.J. Kim

**Affiliations:** 1Department of Naval Architecture and Ocean Engineering, Hongik University, Jochiwon, Sejong 30016, Korea; 2Department of Mechanical Engineering, University College London, London WC1E 7JE, UK

**Keywords:** functionally graded, CNT-reinforced composite, effective stress minimization, exterior penalty-function method, finite difference scheme, thickness-wise CNT distribution

## Abstract

This paper is concerned with the numerical optimization of the thickness-wise CNT (carbon nanotube) distribution in functionally graded CNT-reinforced composite (FG-CNTRC) beams to secure the structural safety. The FG-CNTRC in which CNTs are inserted according to the specific thickness-wise distribution pattern are extensively investigated for high-performance engineering applications. The mechanical behaviors of FG-CNTRC structures are definitely affected by the distribution pattern of CNTs through the thickness. Hence, the tailoring of suitable CNT distribution pattern is an essential subject in the design of FG-CNTRC structure for a given boundary and loading conditions. Nevertheless, the thickness-wise CNT distribution pattern has been assumed by several linear functions so that these assumed primitive patterns cannot appropriately respond to arbitrary loading and boundary conditions. In this context, this paper aims to introduce a numerical method for optimally tailoring the CNT distribution pattern of FG-CNTRC beams. As a preliminary stage, the effective stress is defined as the objective function and the layer-wise CNT volume fractions are chosen as the design variables. The exterior penalty-function method and golden section method are adopted for the optimization formulation, together with finite difference scheme for the design sensitivity analysis. The proposed optimization method is illustrated and validated through the benchmark experiments, such that it successfully provides an optimum CNT distribution which can significantly minimize the effective stress, with a stable and rapid convergence in the iterative optimization process.

## 1. Introduction

Carbon nanotube (CNT) has been spotlighted as a state-of-art material of the 21st century due to its excellent physical, electrical and chemical properties [1,2]. A representative quality is the high weight–stiffness ratio, so that CNTs can be used as a next-generation pillar for polymer-matrix composites. In fact, the mechanical strength of the CNT-reinforced polymer composite dramatically increases when only a small amount of CNTs are inserted [3]. In mechanical applications, CNTRCs have been developed in the form of beams, plates and shells, because these are used as ingredient components in a variety of engineering applications. When compared with the conventional composites, CNTRCs provide superior mechanical behaviors, such as bending deformation, free vibration and buckling [4,5,6].

Meanwhile, the above-mentioned structural components exhibit variations in their mechanical behaviors through their thickness, so that the uniform thickness-wise distribution of CNTs may not be sufficient to respond to such variations. It has been reported that the improvement in the mechanical behavior of CNTRCs is limited when CNTs are uniformly distributed [7,8]. To resolve this limitation, the functional gradient in the thickness-wise CNT distribution was introduced according to the concept of functionally graded material (FGM) [9]. Shen [10] and Ke et al. [11] proposed a purposeful thickness-wise CNT distribution within CNTRCs to suppress bending deformation and control the free vibration. Thereafter, the CNT-reinforced composites with purposeful thickness-wise CNT distributions were called functionally graded CNT-reinforced composites (FG-CNTRCs). The representative FG-CNTRCs include FG-U, FG-V, FG-O and FG-X, in which CNTs are distributed through the thickness with the purposeful linear gradients. The dispersion of CNTs into a polymer matrix can be carried out by the spark plasma-sintering process, the powder metallurgy (PM) route and chemical vapor deposition. The reader may refer to [12,13,14] for more details on the manufacturing of CNT-reinforced composites.

Since their introduction, extensive research efforts have focused on FG-CNTRCs to investigate their mechanical behaviors, particularly the parametric investigation with respect to the thickness-wise distribution pattern of CNTs. For example, Zhu et al. [15] and Cho and Ahn [3] numerically investigated the influence of CNT distribution pattern on the bending deformation and the natural frequencies of FG-CNTRC plates using finite and natural element methods. Cheshmeh et al. [16] investigated the buckling and vibration behaviors of FG-CNTRC plate with respect to the CNT distribution pattern and the CNT volume fraction using the higher-order shear deformation theory. Zhong et al. [17] investigated the free vibration of FG-CNTRC circular and annular plates with arbitrary boundary conditions with respect to the CNT distribution pattern. Xiang et al. [18] numerically investigated the free vibration of FG-CNTRC conical shell panels with respect to the CNT distribution pattern, as well as the key design parameters using the Ritz element-free method. Mirzaei and Kiani [19] analytically investigated the influence of the CNT distribution pattern on the thermal buckling of FG-CNTRC conical shells. Thai et al. [20] proposed a NURBS formulation based on the four-variable refined plate theory for free vibration, buckling and bending analyses of multilayer FG-CNTRC plates. Civalek et al. [21] investigated the free vibration and buckling behaviors of FG-CNT-reinforced cross-ply laminated composite plates using an FSDT and a discrete singular convolution (DSC) method. Mohammadimehr et al. [22] analyzed the buckling and free vibration of double-bonded micro-composite sandwich plates based on the most general strain gradient theories (MGSGT) under electro–magneto–thermo–mechanical and pre-stress loadings. Wu et al. [23,24] investigated the geometrical imperfection sensitivity of postbuckling of FG-CNTRC and piezoelectric FG-CNTRC beams using the first-order shear deformation (FSD) beam theory with von-Kármán nonliearity. Zaghloul et al. [25,26] studied the fatigue and tensile behaviors of polymer composites filled with nanoparticles and presented an experimental and modeling analysis of their mechanical–electrical behaviors. The review papers by Liew et al. [27] and Zaghloul and Zagghloul [14] may be referred to for a broader literature survey of FG-CNTRC structures.

As can be realized from the above literature survey, most studies focused on the parametric investigation of FG-CNTRC structures with respect to the CNT distribution pattern. This parametric study can provide a guideline for choosing a suitable one from the above-mentioned primitive CNT distribution patterns. Nevertheless, this approach can only be called a passive one because the most suitable CNT distribution pattern for the FG-CNTRC under consideration is problem-dependent, as the mechanical behavior of FG-CNTRC is definitely influenced by the structure geometry and the loading/boundary conditions. In this situation, an active approach to tailoring a best thickness-wise CNT distribution pattern, which can maximize the target performance, is required for the successful application of FG-CNTRC structures. Inspired by this situation, the present paper introduces a numerical method for optimally tailoring the thickness-wise CNT distribution pattern for FG-CNTRC structures.

As a preliminary stage, the static bending of FG-CNTRC beam is considered and the optimization problem is defined by minimizing the effective stress. The CNT distribution in the FG-CNTRC beam varies continuously and smoothly throughout the thickness, so the definition of design variables should be considered. In the current work, an FG-CNTRC beam is divided into several sub-layers with uniform CNT distributions in order to minimize the total number of design variables and maintain the flexibility of CNT distribution through the thickness. Then, the CNT volume fractions of each sub-layer become the design variables and their arithmetic sum should satisfy the preset entire CNT volume fraction. The constrained numerical optimization is formulated by employing the exterior penalty-function method, and the sensitivity of objective function to the design variable vector is evaluated using the central difference scheme. The proposed numerical optimization method is validated through the benchmark examples, and the optimum CNT distributions and the associated stress distributions are investigated with respect to the loading condition and compared with those of conventional CNT distribution patterns. The numerical experiments confirmed that the present optimization method successfully seeks the optimum CNT distribution, which provides the minimum effective stress.

The paper is organized as follows: FG-CNTRC structures are introduced in Section 2, together with the CNT volume fraction distribution and the effective material properties. The numerical formulations for the finite-element static analysis and the CNT distribution optimization are addressed in Section 3. The optimization results are presented in Section 4, together with the discussion, and the final conclusion is given in Section 5.

## 2. Modeling of Functionally Graded CNT-Reinforced Composites

Figure 1a shows a typical CNT-reinforced rectangular composite in which single-walled CNTs (SWCNTs) are functionally distributed through the thickness within a polymer matrix. The carbon nanotubes are inserted such that their axes direct to the x–direction, and the composite has the length L, the depth D and the thickness h. The distribution of CNT through the thickness has a functional gradient as shown in Figure 1b, where four types of functionally graded CNTRCs are depicted: FG-U, FG-V, FG-O and FG-X. Since FG-CNTRCs are sort of dual-phase composite, their effective mechanical properties can be determined using the approximate method [9,28] such as the rule of mixture or the Mori–Tanaka estimate. For the current study, the modified linear rule of mixtures (LRM) is used by introducing the CNT efficiency parameters $\eta_{j}\left(j=1,2,3\right)$.

According to the modified LRM, the effective elastic moduli and the effective shear modulus are calculated as [10]
(1)$$E_{1}=\eta_{1}V_{cnt}E_{1}^{cnt}+V_{m}E^{m}$$
(2)$$\frac{\eta_{2}}{E_{2}}=\frac{V_{cnt}}{E_{2}^{cnt}}+\frac{V_{m}}{E^{m}}$$
(3)$$\frac{\eta_{3}}{G_{12}}=\frac{V_{cnt}}{G_{12}^{cnt}}+\frac{V_{m}}{G^{m}}$$
in which $V_{cnt}$ and $V_{m}=1-V_{cnt}$ denote the volume fractions of CNTs and polymer which are in function of z. In Equations (1)–(3), the scripts cnt and m are used to denote the properties of SWCNT and polymer matrix. The polymer matrix is assumed to be homogeneous and isotropic while SWCTs are modeled as an orthotropic material, and, furthermore, it is assumed that $E_{3}=E_{2}$ and $G_{23}^{}=G_{31}^{}=G_{12}^{}$. The scale dependence of the effective material properties of CNTRCs is reflected through the CNT efficiency parameters $\eta_{j}$ [29]. These parameters were obtained by matching the effective CNTRC properties predicted by the molecular dynamics (MD) simulation with those estimated by LRM. Table 1 presents the CNT efficiency parameters for a poly(methyl methacrylate) (PMMA) matrix with CNTs reinforcement at room temperature.

The volume fraction functions of CNTs in the above four FG-CNTRCs through the thickness are expressed by
(4)$$V_{cnt}\left(z\right)=\{\begin{array}{ll} V_{cnt}^{*}, & \;\mathrm{FG}-U\\ \left(1+2z/h\right)V_{cnt}^{*}, & \;\mathrm{FG}-V\\ 2\left(1-2\left|z\right|/h\right)V_{cnt}^{*} & \;\mathrm{FG}-O\\ 2\left(2\left|z\right|/h\right)V_{cnt}^{*} & \;\mathrm{FG}-X \end{array}$$

Here, the total volume fraction $V_{cnt}^{*}$ of CNTs contained within the polymer matrix is calculated from the CNT mass fraction $w_{cnt}$ such that
(5)$$V_{cnt}^{*}=\frac{w_{cnt}}{w_{cnt}+\left(\rho^{cnt}/\rho^{m}\right)-\left(\rho^{cnt}/\rho^{m}\right)w_{cnt}}$$
with $\rho^{cnt}$ and $\rho^{m}$ being the densities of CNTs and the polymer matrix.

Meanwhile, the effective Poisson’s ratio $\nu_{12}$ and the effective density ρ are determined in a similar manner
(6)$$\nu_{\alpha \beta }=V_{cnt}^{*}\nu_{\alpha \beta }^{cnt}+V_{m}\nu_{}^{m},\;\alpha \beta =12\to 23\to 31$$
(7)$$\rho =V_{cnt}\rho_{}^{cnt}+V_{m}\rho_{}^{m}$$

Considering the FG-CNTRCs as a 3-D orthotropic body occupying a bounded domain $\Omega \in {ℜ}^{3}$, its displacement field u(x)=u(x,y,z) under the action of body force f and external load $\hat{t}$ is governed by
(8)$$\sigma_{ij}{\left(u\right)}_{,j}=f_{i}\;in\;\Omega ,\;\;ij=x,y,z$$
with the boundary conditions:(9)$$u=\hat{u}\;on\;\Gamma_{D}$$
(10)$$\sigma_{ij}n_{j}={\hat{t}}_{i}\;on\;\Gamma_{N}$$

Here, $\Gamma_{D}$ and $\Gamma_{N}$ denote the displacement and force boundaries, and $\sigma_{ij}$ and $n_{j}$ are the stress components and the unit normal vector, respectively.

## 3. Analysis and Optimization

### 3.1. Analysis of Bending Deformation

By virtue of the virtual work principle, the previous static equilibrium equations in Section 2 are converted to the following weak form for the elastic deformation field
(11)$$\int_{\;V}\varepsilon_{ij}\left(v\right)\sigma_{ij}^{}\left(u\right)dV=\int_{\;A}v^{T}\hat{t}\;dA$$
for every admissible displacement v. Additionally, using isoparametric finite elements, the actual and virtual displacements u and v are approximated as
(12)$$u=\left[N\right]\left\{\bar{u}\right\},\;v=\left[N\right]\left\{\bar{v}\right\}$$
where [N] is a (3×3N) matrix containing N FE(finite element) basis functions and $\left\{\bar{u}\right\}$ and $\left\{\bar{v}\right\}$ denote the (3N×1) nodal vectors.

Next, two matrices [D] and [E] are introduced to define the strain vector $\left\{\varepsilon_{ij}\right\}$ and the stress vector $\left\{\sigma_{ij}\right\}$, respectively
(13)$$\left[D\right]=\left[\begin{array}{cccccc} \partial /\partial x & 0 & 0 & \partial /\partial y & 0 & \partial /\partial z\\ 0 & \partial /\partial y & 0 & \partial /\partial x & \partial /\partial z & 0\\ 0 & 0 & \partial /\partial z & 0 & \partial /\partial y & \partial /\partial z \end{array}\right]$$
(14)$$\left[E\right]=diag\left[E_{1,}E_{2},E_{3},G_{12},G_{23},G_{31}\right]$$

The components in [E] denote the effective orthotropic material properties of CNTRC structures, which will be given later. Using these two matrices, together with the FE approximation (12), both strains and stresses are approximated as
(15)$$\left\{\varepsilon_{ij}\left(v\right)\right\}=\left[H\right]\left\{\bar{v}\right\},\left[H\right]=\left[D\right]\left[N\right]$$
and
(16)$$\left\{\sigma_{ij}^{}\left(u\right)\right\}=\left[E\right]\left[H\right]\left\{\bar{u}\right\}\;$$

Introducing Equations (15) and (16) into the weak form (11), one can obtain the simultaneous linear equations to solve the static deformation problem:(17)$$\left[K\right]\left\{\bar{u}\right\}=\left\{f\right\}$$
where the stiffness matrix [K] and the load vector {f} are defined by
(18)$$\left[K\right]=\int_{\;V}\left[H^{T}\right]\left[E\right]\left[H\right]\;dV$$
(19)$$\left\{f\right\}=\int_{\;A}\left\{N^{T}\right\}\hat{t}\;dA$$

### 3.2. Optimization of CNT Distribution

For the present numerical optimization, an FG-CNTRC beam with the specific thickness-wise CNT gradient is divided into (ND) numbers of uniform homogenized sub-layers, as depicted in Figure 2. Then, the CNT volume fractions ${\left(V_{cnt}\right)}_{i}$ of each sub-layer define the design variable vector X:(20)$$X=\left\{{\left(V_{cnt}\right)}_{1},\;{\left(V_{cnt}\right)}_{2},\;\ldots ,\;{\left(V_{cnt}\right)}_{ND}\right\}$$

From the physical constraint, each layer-wise CNT volume fraction ${\left(V_{cnt}\right)}_{i}$ must satisfy the following upper and lower bounds
(21)$$0\leq {\left(V_{cnt}\right)}_{i}\leq 1,\;i=1,\;2,\ldots ,\;ND$$
and their sum should be equal to the preset volume fraction $V_{cnt}^{*}$ of CNTs:(22)$${\left(V_{cnt}\right)}_{1}+{\left(V_{cnt}\right)}_{2}+\cdots +{\left(V_{cnt}\right)}_{ND}=ND\times V_{cnt}^{*}$$

Next, the peak effective stress is defined as the objective function F(X), such that
(23)$$F\left(X\right)=\underset{x\in \Omega }{\max}\left|\sigma_{eff}\left(x;X\right)\right|$$
with Ω being the entire material domain of FG-CNTRC beam.

Then, the constrained optimization of thickness-wise CNT distribution is formulated as follows, together with the FE approximation (17):(24)$$\mathrm{Find}\;\;X={\left\{X_{i}\right\}}_{i=1}^{ND},\;X_{i}={\left(V_{cnt}\right)}_{i}$$
(25)Minimize F(X)
(26)$$\mathrm{Subject}\;\mathrm{to}\;\left[K\right]\left\{\bar{u}\right\}=\left\{f\right\}$$
(27)$$h\left(X\right):\sum_{i=1}^{ND}{\left(V_{cnt}\right)}_{i}-ND\times V_{cnt}^{*}=0$$
(28)$$g_{j}\left(X_{j}\right):-{\left(V_{cnt}\right)}_{j}\leq 0,\;j=1,\;2,\ldots ,\;ND$$
(29)$$g_{j}\left(X_{j-ND}\right):{\left(V_{cnt}\right)}_{j-ND}-1\leq 0,\;j=ND+1,\ldots ,\;2*ND$$

Note that Equation (27) is the equality constraint, while Equations (28) and (29) are the inequality constraints.

The constrained optimization problem is solved by employing the exterior penalty-function method (EPFM) [30]. This method converts the constrained objective function F(X) to an unconstrained pseudo-objective function $\Phi \left(X;r_{p}\right)$, by adopting the exterior penalty parameters $r_{p}$, such that
(30)$$\Phi \left(X;r_{p}\right)=F\left(X\right)+r_{p}c_{1}\left\{h^{2}\left(X\right)\right\}+r_{p}c_{2}\sum_{j=1}^{2*ND}\max\left[0,g_{j}^{2}\left(X\right)\right]$$

Meanwhile, the maximum value of objective function is in the order of $10^{6}\sim 10^{7}$ while those of the constraints are 1.0, as will be shown later. Therefore, the normalization factors $c_{1}$ and $c_{2}$ are inserted to maintain the balance between the magnitudes of constraints and objective function. These two factors are calculated using the usual normalization given by
(31)$$c_{1}=\frac{\max\;\left|\nabla F\left(X\right)\right|}{\max\;\left|\nabla h\left(X\right)\right|},\;\;c_{2}=\frac{\max\;\left|\nabla F\left(X\right)\right|}{\underset{j}{\max}\;\left|\nabla g_{j}\left(X_{j}\right)\right|}$$

The definition of inequality constraints given in Equations (28) and (29) leads to
(32)$$\begin{array}{ll} \left|\nabla g_{j}\left(X_{\ell }\right)\right| & =\left|\partial g_{j}/\partial X_{\ell }\right|\\ & =\left|\pm \delta_{j\ell }\right|\leq 1 \end{array}$$

Plugging Equation (32) into Equation (31) results in
(33)$$c_{2}=\max\;\left|\nabla F\left(X\right)\right|$$

The iterative optimization process starts with an initial design variable $X^{0}$. At each iteration step k (k=1, 2,…), the sensitivity analysis presented in Section 3.3 is performed and the convergence is examined as follows
(34)$$\left|F\left(X^{k}\right)-F\left(X^{k-1}\right)\right|/\left|F\left(X^{k}\right)\right|\leq \varepsilon_{T}$$
with $\varepsilon_{T}$ being the convergence tolerance. The iterative optimization is terminated when the convergence criterion is satisfied; otherwise, the optimization process proceeds to the next iteration by updating the current exterior penalty parameter
(35)$$r_{p}^{k+1}=\gamma \cdot r_{p}^{k}$$

Here, γ (γ>1) indicates an iteration-independent update constant, which successively increases the penalty parameter during the iterative optimization. The flowchart of the proposed optimization process is represented in Figure 3.

### 3.3. Sensitivity Analysis

The sensitivity analysis calculates the direction vector S of the design variable, which is essential for searching the optimization direction. From Equations (23) and (27)–(29), the sensitivity of the pseudo-objective function $\Phi \left(X;r_{p}\right)$ to the *i*th design variable $X_{i}$ is expressed by
(36)$$\frac{\partial \Phi \left(X;\;c,\;r_{p}\right)}{\partial X_{i}}=\frac{\partial \left\{\underset{x\in \Omega }{\max}\left|\sigma_{eff}\left(x;X\right)\right|\right\}}{\partial X_{i}}+2c_{1}r_{p}h\left(X\right)+2c_{2}r_{p}\left[{\left(V_{cnt}\right)}_{j}-1\right],i=1,\;2,\;\ldots ,ND$$

This direct method may be considered when the thickness-wise CNT distribution is assumed a priori such that the objective function F(X) is expressed in an explicit form of CNT volume fractions ${\left(V_{cnt}\right)}_{i}$. However, it can be assumed that the first term on the right-hand side of Equation (36) requires a complex and painstaking analytical derivation.

An effective and attractive alternative is to employ the finite difference method because the current optimization problem is not large-scale. When the finite difference is used, the direction vector $S^{k}=\left\{S_{1}^{k},\;S_{2}^{k},\;\ldots ,\;S_{ND}^{k}\right\}$ at the *k*th stage is computed by
(37)$$S^{k}=\frac{\Phi \left(X^{k-1}+\delta X;\;c_{1}^{k-1},\;c_{2}^{k-1},\;r_{p}^{k-1}\right)-\Phi \left(X^{k-1};\;c_{1}^{k-1},c_{2}^{k-1},r_{p}^{k-1}\right)}{\delta X},k=1,\;2,\;\ldots$$
with $r_{p}^{0}$ being the initial exterior penalty parameter. When the direction vector is obtained, the design variable is updated such that
(38)$$X^{k}=X^{k-1}+\Delta X^{k},\Delta X^{k}=\beta^{k}S^{k}$$
with the iteration-dependent constants $\beta^{k}$ for determining the direction vector magnitude.

Two representative methods are widely used to decide the magnitude of direction vector: Lagrange interpolation and golden section methods [30]. Both methods commonly seek the critical value of β that minimizes the pseudo-objective function $\Phi \left(X;\;c,\;r_{p}\right)$, but the latter is preferable because it always secures the local minimum to the design variable vector. The detailed description for this method is skipped because the numerical implementation of this method is standardized.

## 4. Results and Discussion

Figure 4a represents the first model problem, a simpply supported FG-CNTRC beam subject to uniform distributed load q=0.1 MPa. The length L is 0.1m and the depth D and the thickness h are equally set by 0.01m, respectively. The matrix is manufactured with poly(methyl methacrylate) (PMMA) and its isotropic material properties are given in Table 2, where the orthotropic material properties of the (10,10) SWCNTs are also presented. The FG-CNTRC beam is divided into 10 uniform sub-layers with layer-wise CNT volume fractions ${\left(V_{cnt}\right)}_{i}\left(i=1,2,\cdots ,10\right)$, as represented in Figure 4b, in order to suppress the increase in the total number of design variables by considering the flexibility of thickness-wise CNT distribution at the same time. For the finite element structural analysis, each sub-layer is discretized into 100×10 uniform 8-node cubic elements, with 100 in the x-direction and 10 in the y-direction. Therefore, the total number of finite elements for the whole FG-CNTRC beam reaches 10,000.

The initial CNT volume fractions ${\left(V_{cnt}\right)}_{i}$ of ten sub-layers are set by the preset total CNT volume fraction $V_{cnt}^{*}$. For example, the initial layer-wise CNT volume fractions ${\left(V_{cnt}\right)}_{i}$ are equally designated by 0.12 when the preset value of $V_{cnt}^{*}$ is 0.12. The convergence tolerance $\varepsilon_{T}$ in Equation (34) is set by $1.0\times 10^{-3}$ and the initial penalty parameter $r_{p}^{0}$ and the update constant γ in Equation (35) are taken by 1.0 and 2.0, respectively. Since this problem exhibits a remarkable edge effect in the bending stress field near the left and right ends of beam, the peak stress value is examined from the thickness-wise stress distribution at the beam mid-span (i.e., at x=L/2). The finite element analysis was carried out by midas NFX [31], a commercial FEM software.

For the preset CNT volume fraction $V_{cnt}^{*}=0.12$, the optimization process terminates after five iterations, as shown in Table 3. The peak effective stress occurs at the top and bottom of beam, and 311 FEM analyses were performed, mostly for the sensitivity analysis. Table 3 shows that the objective function uniformly decreases proportionally to the iteration number. The peak effective stress $\sigma_{eff}^{\max}$ is 7.504 MPa at the initial stage and 5.947 MPa at the final stage, so that it is reduced by 1.557 MPa, which corresponds to 20.7% of the initial peak effective stress. Figure 5a compares the initial and optimum CNT distributions, where the layer-wise CNT volume fractions ${\left(V_{cnt}\right)}_{i}$ are larger than 0 and less than 1.0 and their sum is found to be 11.95. Hence, it is confirmed that the equality constraint in Equation (27) and the inequality constraints in Equations (28) and (29) are strictly enforced. Figure 5b compares the thickness-wise effective stress distributions between the initial and optimum CNT distributions, where the initial one varies linearly from zero at the mid-surface to the peak value at the top and bottom. The optimum one is also zero at the mid-surface and symmetric with respect to the mid-surface, but is not linear any longer and its peak is smaller than that of the initial one. In the bending deformation, the bending strain exhibits a linear thickness-wise variation, which produces the linear bending stress distribution through the thickness when the elastic modulus is uniform. However, the parabolic-type optimum CNT distribution leads to the parabolic-type distribution of elastic modulus of FG-CNTRC beam, so that the non-linear stress distribution with the smaller peak effective stress is revealed.

The optimization is also performed for two different CNT volume fractions, $V_{cnt}^{*}=0.17$ and 0.28, and the optimization results are compared in Table 4. The total iteration numbers are similar, but the total numbers of FE analyses are different. It is presumed that the sensitivity analysis is influenced by the value of $V_{cnt}^{*}$. The peak effective stresses occur at the top and bottom of beam for all the three cases, and the initial and optimum peak effective stresses are similar. Thus, it is found that the influence of the total CNT volume fraction $V_{cnt}^{*}$ on the CNT distribution optimization for minimizing the peak effective stress is not significant.

Figure 6a comparatively represents the optimum CNT distributions for three different values of $V_{cnt}^{*}$. It can be seen that three optimum CNT distributions are commonly parabolic-type and show different layer-wise CNT volume fractions ${\left(V_{cnt}\right)}_{i}$ to fulfill the equality constraint imposed upon $V_{cnt}^{*}$. Figure 6b compares the effective stress distributions between three different values of $V_{cnt}^{*}$, where it is observed that the difference between three distributions is not significant. Thus, it is again confirmedthat the total CNT volume fraction $V_{cnt}^{*}$ does not impose any significant influence on the stress results.

Next, the thickness-wise stress distribution for the optimum CNT distribution when $V_{cnt}^{*}$ is 0.12 is compared with those of four standard functionally graded CNT distributions, FG-U, FG-V, FG-O and FG-X. The results are presented in Table 5, where $\Delta \sigma_{eff}^{\max}$ indicates the difference in $\sigma_{eff}^{\max}$ between the optimum and FG CNT distributions and the values within parenthesis denote the relative differences with respect to the optimum stress. It is found that $\sigma_{eff}^{\max}$ is lowest for the optimum case, while this is the highest for FG-V. Thus, it has been justified that the present optimization method successfully seeks the optimum CNT distribution, which can provide the maximum effective stress, which is smaller than those of standard FG CNT distributions.

This fact is well represented in Figure 7a, where FG-V shows an unsymmetric effective stress distribution with the highest $\sigma_{eff}^{\max}$ at the top of the beam because its CNT distribution is unsymmetric, as shown in Figure 1b. Meanwhile, FG-O produces the peak effective stress at $\bar{z}=\pm 0.3$ in accordance with its CNT distribution. Furthermore, its peak effective stress is higher than that of FG-U even though its CNT distribution is similar to the optimum CNT distribution. For the sake of comparison, the axial stress distributions are comparatively represented in Figure 7b, to examine whether there is a remarkable difference between the effective stress, which is calculated by all six stress components, and a single bending stress component. Since the axial stress is the dominant stress component in the current bending deformation problem, the axial stress distributions are observed to be similar to the effective stress distributions, except for the minus sign in the upper-half region.

Next, the optimization was performed again for the previous simply supported beam by inclining the distributed load. Referring to Figure 4a, the vertical distributed load was inclined by α=45° with respect to the z-axis. However, except for this inclined distributed load, the beam geometry and the simulation parameters were kept the same as the previous case. The layer-wise CNT volume fractions ${\left(V_{cnt}\right)}_{i}$ were initially set by 0.12, and the numerical optimization terminates in five iterations, as presented in Table 6. When compared with the previous case, the total number of FEM analyses was smaller and the location of peak effective stress was fixed at the bottom. The peak effective stress $\sigma_{eff}^{\max}$ ws 5.666 MPa at the initial stage and 4.321 MPa at the final stage, being reduced by 1.345 MPa, which is 23.7% of the initial peak effective stress.

Figure 8a compares the iteration histories of objective function between the previous and current cases, where both cases produce a difference in effective stress magnitude but commonly show rapid and stable convergence. Figure 8b comparatively represents the initial and optimum CNT distributions, where the arithmetic sum of layer-wise CNT volume fractions ${\left(V_{cnt}\right)}_{i}$ was found to be 12.08. Thus, the preset CNT volume fraction $V_{cnt}^{*}=0.12$ is strictly satisfied. Meanwhile, the current optimum CNT distribution was seen to be asymmetric, differing from the previous one shown in Figure 5a. This is because the inclined distributed load produces the asymmetric effective stress distribution, as will be seen below. Thus, it has been found that the optimum thickness-wise CNT distribution is significantly influenced by the loading condition. Additionally, this asymmetric CNT distribution cannot be tailored, even though the conventional primitive CNT distribution patterns shown in Figure 1b are combined.

Figure 9a comparatively represents the axial stress distributions of the initial and optimum CNT distributions. Differing from the previous case shown in Figure 7b, the axial stress in the present case does not vanish at the mid-surface. In other words, the zero-line moves slightly upwards because the inclined load not only produces the bending deformation but also the axial extension. Thus, the initial axial stress distribution is not symmetric but asymmetric with respect to the mid-surface. The optimum axial stress distribution becomes more asymmetric because the optimum CNT distribution is additionally asymmetric, as shown in Figure 8b. Figure 9b compares the thickness-wise effective stress distributions between the initial and optimum CNT distributions, where both the initial and optimum CNT distributions show higher peak effective stresses at the bottom because the inclined distributed load produces additional tensile stress. When compared with the previous case shown in Figure 5b, the present inclined load leads to a remarkably different effective stress distribution. The effective stress does not vanish near the mid-surface because the shear stress caused by the inclined distributed load is not negligible. Hence, it is found that not only the optimum CNT distribution, but also the resulting stress distribution, is strongly influenced by the loading condition.

## 5. Conclusions

A numerical optimization method was introduced to seek the optimum thickness-wise CNT distribution that minimizes the effective stress in FG-CNTRC beams. An FG-CNTRC beam with a continuous CNT distribution was divided into several sub-layers with uniform CNT distributions, and the optimum CNT volume fractions of each sub-layer were sought by the exterior penalty-function method. This approach not only reduced the total number of design variables but simplified the numerical formulation. The benchmark numerical experiments were conducted to illustrate and validate the proposed numerical method and investigate the optimization results. Through the numerical result, the following major observations can be drawn:The proposed method successfully seeks the optimum thickness-wise CNT distribution, which minimizes the effective stress, with rapid and stable convergence.For the vertical distributed load, the optimum CNT distribution is symmetric and parabolic, and the effective stress distribution is also symmetric.The peak effective stress occurred at the top and bottom, and was reduced by 20.7% after the optimization.The optimum CNT distribution is different from the four primitive CNT distributions and leads to the peak effective stress, which is reduced by at least 26.2% compared to those of four primitive patterns.Both the optimum CNT distribution and the associated stress distributions are significantly influenced by the loading condition.For the inclined distributed load, both distributions become un-symmetric, but the initial peak stress that occurred at the bottom is reduced by 23.7% after optimization.

The proposed optimization method can be practically used to tailor the thickness-wise CNT distribution and secure the structural safety of FG-CNTRC beams against the external load. The current work is limited to the minimization of effective stress, so the optimized CNT distribution pattern may lead to a reduced structural sfiffness and, consequently, a larger bending deflection. Hence, the trade-off between the effective stress and the bending deflection would be worthwhile, and is a deserving topic for future work.

## Figures and Tables

**Figure 1 polymers-14-04418-f001:**
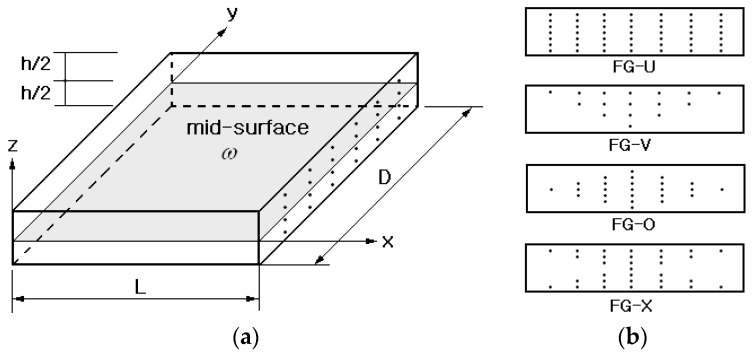
A carbon-nanotube-reinforced composite (CNTRC): (**a**) geometry and dimensions; (**b**) four functionally graded (FG) distributions.

**Figure 2 polymers-14-04418-f002:**
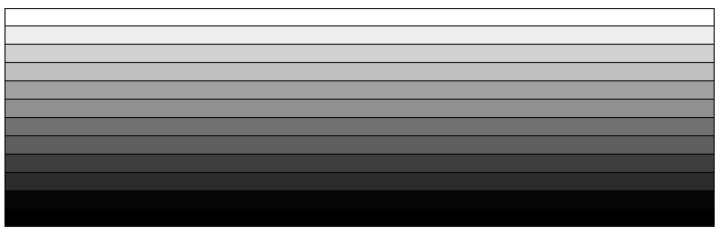
Division of an FG-CNTRC beam into uniform homogenized sub-layers.

**Figure 3 polymers-14-04418-f003:**
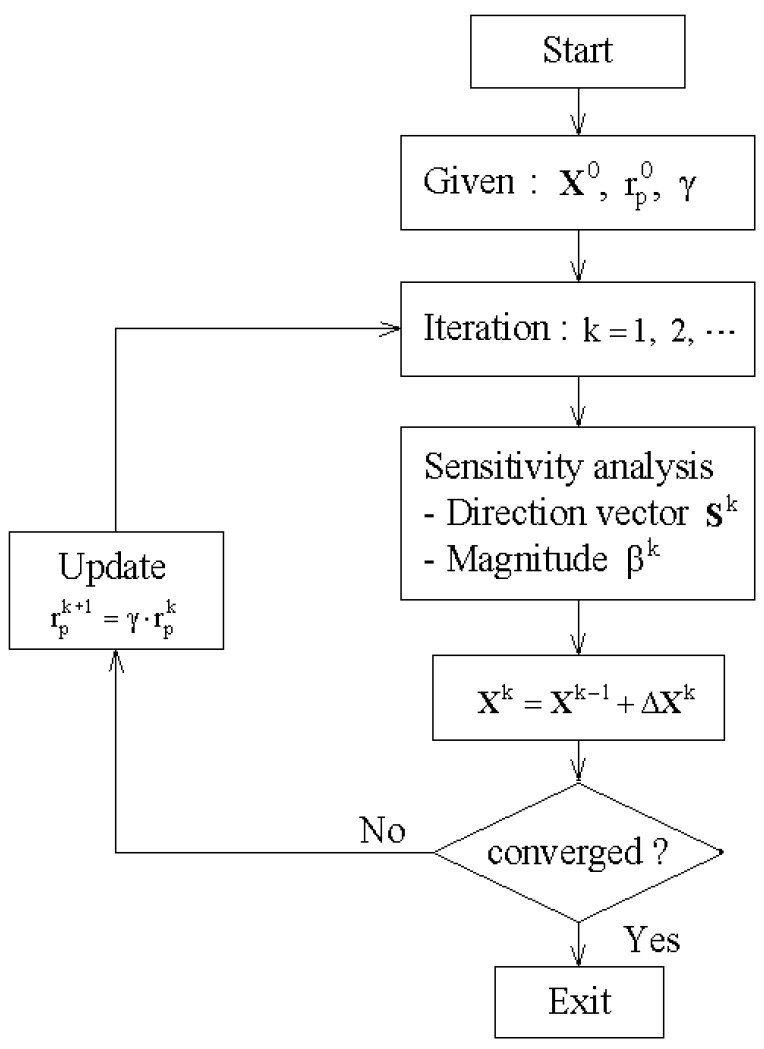
Flowchart of the CNT distribution optimization.

**Figure 4 polymers-14-04418-f004:**
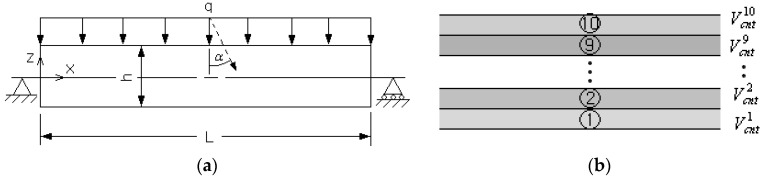
A simply supported FG-CNTRC beam: (**a**) geometry and dimensions, (**b**) uniform sub-layers.

**Figure 5 polymers-14-04418-f005:**
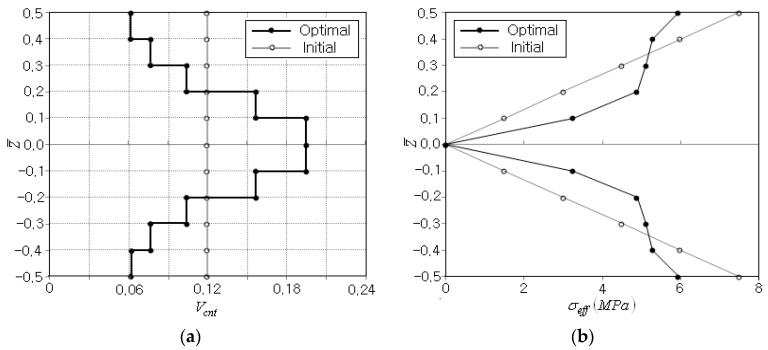
Comparison between the initial and final distributions: (**a**) the CNT volume fraction $V_{cnt}^{*}$, (**b**) the effective stress $\sigma_{eff}$.

**Figure 6 polymers-14-04418-f006:**
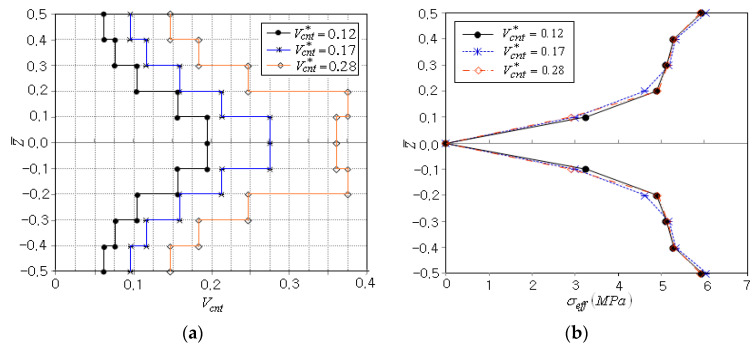
Comparison between the volume fractions $V_{cnt}^{*}$: (**a**) CNT distribution, (**b**) effective stress distribution $\sigma_{eff}$.

**Figure 7 polymers-14-04418-f007:**
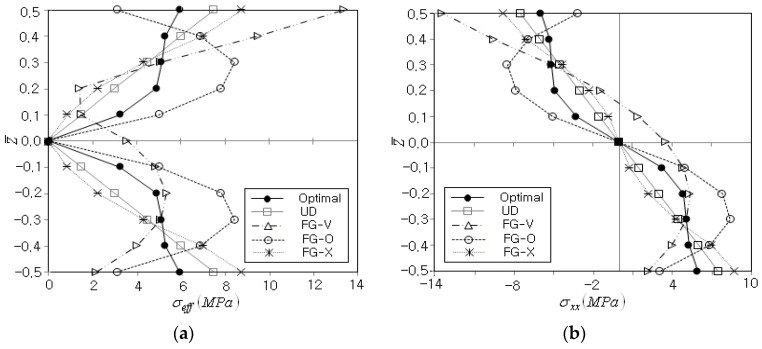
Comparison of thickness-wise stress distributions ($V_{cnt}^{*}=0.12$): (**a**) the effective stress, (**b**) the axial stress.

**Figure 8 polymers-14-04418-f008:**
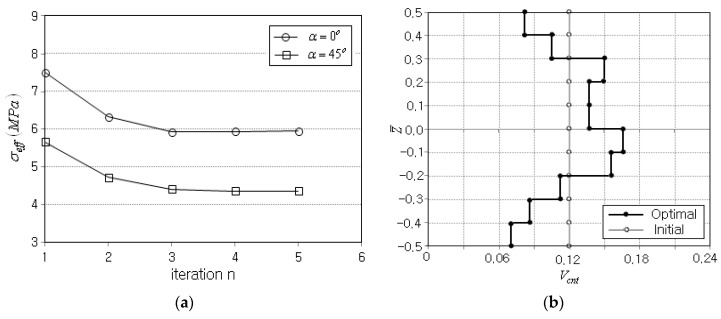
Comparison (α=45°): (**a**) the variations in objective function, (**b**) the distributions of initial and optimum CNT volume fractions $V_{cnt}^{*}$.

**Figure 9 polymers-14-04418-f009:**
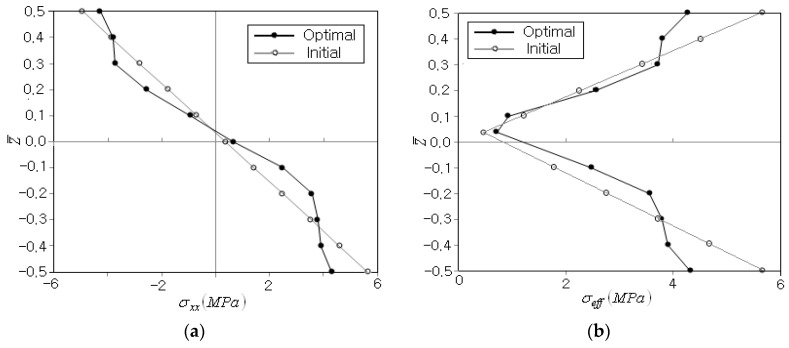
Comparison of thickness-wise stress distributions (α=45°): (**a**) the axial stress, (**b**) the effective stress.

**Table 1 polymers-14-04418-t001:** The CNT efficiency parameters for three different $V_{cnt}^{*}$ (PMMA/CNT at T=300K).

$V_{cnt}^{*}$	$\eta_{1}$	$\eta_{2}$	$\eta_{3}$
0.12	0.137	1.022	0.715
0.17	0.142	1.626	1.138
0.28	0.141	1.585	1.109

**Table 2 polymers-14-04418-t002:** Material properties of SWCNT and matrix (PMMA).

Materials	Young’s Modulus (*GPa*)			Poisson’s Ratio			Shear Modulus *GPa*			Density $\left(kg/m^{3}\right)$
	$E_{1}$	$E_{2}$	$E_{3}$	$\nu_{12}$	$\nu_{23}$	$\nu_{31}$	$G_{12}$	$G_{23}$	$G_{31}$	ρ
SWCNT	5646.6	7080.0	-	0.175	-	-	1944.5	-	-	1400
PMMA	2.5			0.34			0.9328			1150

**Table 3 polymers-14-04418-t003:** The iteration history of objective function.

Iteration	Objective Function $\sigma_{eff}^{\max}$	Location (z)
Initial	7.50387 $\times \;10^{6}\;\mathrm{Pa}$	±5.0
1	6.32390 $\times \;10^{6}\;\mathrm{Pa}$	±5.0
2	5.93291 $\times \;10^{6}\;\mathrm{Pa}$	±5.0
3	5.94156 $\times \;10^{6}\;\mathrm{Pa}$	±5.0
4	5.93291 $\times \;10^{6}\;\mathrm{Pa}$	±5.0
Total number of FEM analyses		311

**Table 4 polymers-14-04418-t004:** The optimization results for three different total CNT volume fractions $V_{cnt}^{*}$.

Items	CNT Volume Fractions $V_{cnt}^{*}$		
	0.12	0.17	0.28
Initial, $\sigma_{eff}^{\max}\left(X_{0}\right)\;\left(MPa\right)$	7.50387	7.50383	7.50366
Optimum, $\sigma_{eff}^{\max}\left(X_{opt}\right)\;\left(MPa\right)$	5.94722	6.04832	5.91510
Iterations	5	4	5
Total number of FEM analyses	311	164	270

**Table 5 polymers-14-04418-t005:** Comparison of the maximum effective stresses between the optimum and conventional CNT distributions.

Items	CNT Distribution				
	Optimum	FG-U	FG-V	FG-O	FG-X
$\sigma_{eff}^{\max}\;\left(MPa\right)$	5.94722	7.50365	13.39331	8.45360	8.76545
$\Delta \sigma_{eff}^{\max}\;\left(MPa\right)$	-	1.55643 (26.2%)	7.44609 (125.2%)	2.50638 (42.1%)	2.81823 (47.3%)
Location $\left(\bar{z}\right)$	±0.5	±0.5	+0.5	±0.3	±0.5

**Table 6 polymers-14-04418-t006:** The iteration history of objective function (α=45°).

Iteration	Objective Function $\sigma_{eff}^{\max}$	z
Initial	5.66564 $\times \;10^{6}\;\mathrm{Pa}$	−5.0
2	4.27414 $\times \;10^{6}\;\mathrm{Pa}$	−5.0
3	4.21625 $\times \;10^{6}\;\mathrm{Pa}$	−5.0
4	4.35744 $\times \;10^{6}\;\mathrm{Pa}$	−5.0
5	4.32146 $\times \;10^{6}\;\mathrm{Pa}$	−5.0
Total number of FEM analyses		186

## Data Availability

Not applicable.

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
