# Peer review of "Numerical Optimization of CNT Distribution in Functionally Graded CNT-Reinforced Composite Beams"

_polymers, 2022, doi:10.3390/polym14204418_

Round 1
Reviewer 1 Report
In the present article, a numerical method for optimally tailoring the carbon nanotube distribution pattern of functionally graded carbon nanotube reinforced beams. The effective stress is defined by the objective function and the layer-wise CNT volume fractions are chosen as the design variables.
The paper is well written and adding a good amount of knowledge to the current field and well within the journal scope. The paper can be accepted after a few small modifications as listed below:
1) The novelty of this work must be more explained.
2) In general, the paper needs to check for proofreading.
3) Why classsical plate theory used than higher-order mid-plane kinematics .
4) The introduction should be improved by considering recent related works.
Free vibration, buckling, and
bending analyses of multilayer functionally graded graphene nanoplatelets reinforced composite plates using the NURBS formulation. Composite Structures 2019:220: 749-59.
Frequencies of FGM shells and annular plates by the methods of discrete singular convolution and differential quadrature methods. Composite Structures 183, 7-20 (2018).
Buckling and free vibrations of CNT-reinforced cross-ply laminated composite plates. Mechanics Based Design of Structures and Machines 50 (6), 1914-1931 (2022)
Buckling and vibration analyses of MGSGT double-bonded micro composite sandwich SSDT plates reinforced by CNTs and BNNTs with isotropic foam & flexible transversely orthotropic cores. Structural Engineering and Mechanics 2018:65: 491-504.
5) The references in the introduction are appropriate but inadequate. Therefore, the introduction should be expanded by adding the above articles to the introduction.
6) All manuscript should be checked carefully for any further typos. Also, The authors are encouraged to show the main practical applications.
Author Response
Please refer to the attached response to reviewers' comments (1).

Reviewer 2 Report
This paper introduces a numerical method for optimally tailoring the CNT distributon pattern so as to obtain the lowest effective stress for a bending problem of a simply supported beam. The paper is overall well written and could be accepted for publication if the following issues are well addressed:
(1) The literature review could be more comprehensive by mentioning some highly relevant works on FG-CNTRCs, such as Applied Mathematical Modelling 42, 735-752; Thin-Walled Structures 108, 225-233; Smart Materials and Structures 25 (9), 095022.
(2) Abbreviations should be expanded at the first use, such as RHS in page 7.
(3) How is the effective stress defined, or say what is the relation between the effective stress and stress components along axes?
(4) The optimized results lack validation against the published results or FEM. This is very important for convincing the readers
(5) Why is only the effective stress selected as the objective function? As far as the reviwer knows, the optimized CNT distribution pattern (similar to FGO) may lead to a reduced stiffness and consequently a larger bending deflection. How to balance the low stress and large deflection?
(6) There are some typo errors, for example, line 207 in page 7, Fig.5 should read as Fig.3.
Author Response
Please refer to the attached response to reviewers' comments (2).

Reviewer 3 Report
The paper presents an interesting approach based on the Numerical Optimization of CNT Distribution in Functionally Graded CNT-reinforced Composite Beams. However, the innovation of the current research work should be further highlighted and emphasized. At the same time, the authors should consider the following comments to greatly improve the quality of the paper.
1. In the abstract, introduce the problem in the initial lines of the abstract.
2. The introduction needs to be improved by relating to the mechanics of the studied materials and their mechanical characteristics. The references to be included are: 10.1177/0021998318790093, 10.1016/j.polymertesting.2017.09.009, 10.1016/j.compstruct.2021.114698, 10.1177/0731684417727143, 10.1002/app.46770, 10.1016/j.porgcoat.2022.107015.
3. Why 10 sub-layers were used for representing the CNT particles?
4. Was there a reason to use SWCNT instead of MWCNT?
5. Which type of CNT was referred to in the model? There are different types of CNT that vary in physical and chemical properties.
6. The dispersion of CNT into PMMA needs further explanation from experimental and validation point of views.
7. The conclusion needs to be modified to summarize the research outcomes in short statements with clear observations.
Author Response
Please refer to the attached response to reviewers' comments (3).

Round 2
Reviewer 1 Report
Required corrections have been made.
Reviewer 2 Report
I am satified with the revision.
Reviewer 3 Report
The comments have been successfully done. The authors are encouraged to add these references in the introduction:
10.1177/0731684417727143
10.1002/app.46770,
10.1016/j.porgcoat.2022.107015.